# SMM Transformer: Leveraging Spiking Neural Networks for Multimodal Tasks

Xiubo Liang [1]   Jinxing Han [1]   Yuke Li [2]   Haoqi Zhu [2]   Yu Zhao [1]   Hongzhi Wang [1 2]

## Abstract

Spiking Neural Networks (SNNs) enable event-driven computation with sparse activations, but building multimodal Transformers on SNNs is hindered by unstable training in deep spiking stacks and the mismatch between dense softmax attention and spike-based communication. We propose SMM Transformer, an SNN-based multimodal Transformer framework that combines (i)PLMP, a Parallel LIF with Multistage Learnable Parameters neuron and a tailored P-STBP algorithm for stable deep SNN training, (ii) SMSA, an attention-inspired spike-driven token-mixing module that replaces dense pairwise softmax attention with channel-wise spike co-activation and self-compensation, and (iii)SMoE, a spiking mixture-of-experts module for modality-aware fusion. Across visual and multimodal benchmarks, SMM Transformer achieves competitive accuracy compared to ANN baselines. Under a standard MAC/AC arithmetic model, SMSA reduces the estimated operator-level compute energy of the attention module by up to 97%, while whole-model profiling shows more moderate but consistent efficiency gains.

## 1. Introduction

Large language models (LLMs) and multimodal foundation models have rapidly advanced vision–language learning, yet their training and deployment costs motivate research into more energy-efficient alternatives (Kasneci et al., 2023; Wei et al., 2022; Carlini et al., 2021; Dubey et al., 2024). In this work, we do not aim to scale to LLM-sized architectures; instead, we target the dominant compute bottleneck in Transformer-style multimodal systems—dense attention—and study how event-driven spiking computation can reduce its operator-level energy cost. By contrast, spiking neural

networks (SNNs) emulate neuronal firing with sparse binary spikes and offer a promising substrate for energy-efficient computing research (Roy et al., 2019).

Multimodal learning underpins assistive vision and large-scale media annotation. Modern vision–language models typically rely on dense Transformer attention to align image and text representations, which is accurate but compute intensive (Xu et al., 2016; Aafaq et al., 2019; Ren et al., 2015; Yang et al., 2016; Lei et al., 2018; Xiao et al., 2022). On the SNN side, prior work has largely focused on single-modality perception and spiking backbones (Zhou et al., 2022), while multimodal vision–language SNNs remain underexplored and often rely on separate encoders or late fusion that limits early cross-modal interaction (Wang et al., 2025; Marković et al., 2020; Furber, 2016; van De Burgt et al., 2018). These gaps motivate a Transformer-inspired spiking architecture that supports efficient fusion under sparse spiking computation.

This study explores Transformer–SNN hybrids for high-accuracy and energy-aware multimodal learning. To mitigate gradient attenuation and tedious parameter tuning in LIF-based SNNs (Tavanaei et al., 2019), we introduce the PLMP neuron. PLMP uses multiple learnable LIF branches to capture heterogeneous temporal dynamics, followed by a fixed-threshold read-out gate that produces a binary spike. We further tailor Spatio-Temporal Back-propagation (STBP) (Wu et al., 2018) into P-STBP to propagate gradients through the multi-branch temporal dynamics and the read-out gate.

For token mixing, we propose SMSA, a PLMP-based spike-driven module inspired by self-attention. SMSA replaces the dense $N \times N$ token-pair matrix with spike co-activation, sequence-level channel aggregation, value masking, and a lightweight self-compensation branch. This design trades part of the dense token-to-token inductive bias for sparse, event-driven computation. For multimodal fusion, we introduce a Spiking Mixture-of-Experts (SMoE), which allocates modality-specific feed-forward capacity to vision, language, and vision–language experts while maintaining a shared interaction pathway.

We retain the Transformer macro-architecture with stacked residual blocks, but redesign the spiking neuron, token-mixing, and feed-forward fusion components to be com-

[1]School of Software Technology, Zhejiang University, Ningbo, China [2]NetEase Yidun AI Lab, Hangzhou, China. Correspondence to: Hongzhi Wang <hongzhiwang@zju.edu.cn>.

*Proceedings of the 43$^{rd}$ International Conference on Machine Learning*, Seoul, South Korea. PMLR 306, 2026. Copyright 2026 by the author(s).

| Image | |  |  |  |  |
|---|---|---|---|---|---|
| **Caption** | Fusecap | A picture of a bowl of orange juice sits on a wooden table next to a white laptop, with a white napkin nearby. | A picture of a traffic light with a red light and a black arrow pointing to the left, next to a white sign with the text stop and a blue sky in the background. | A picture of a group of people stand in front of two large tanks, one green and one black, in front of tall buildings under a clear blue sky is a white sign. | A picture of a stone arch stands tall amidst a lush green landscape, with a blue flag waving in the breeze under a clear blue sky. |
| | BLIP-Large | There is a glass of orange juice on a napkin next to a laptop. | There is a traffic light with a sign on it that says only. | There is a tank that is sitting on the street with people standing on top. | Arafed arch with flags on top of it in a city. |
| | GIT | A cup of orange juice sitting next to a laptop. | A traffic light with a left turn only sign on it. | A tank with people on top of it. | The triumphal arch is made of stone. |
| | Ours | A glass of orange juice rests on a wooden table beside a white laptop, accompanied by a white napkin. | A traffic light displaying a red light and a "no left turn" sign is set against a blue sky. | A group of people stands on a large tank, set against tall buildings under a clear blue sky. | A stone arch with a flag stands before a building, with two people nearby. |
| Image | |  |  |  |  |
| **Caption** | Fusecap | A picture of a group of people, including a man in a blue shirt and jeans, stand in a field holding a black camera. | A picture of a man sits on a pink sailboat with a blue and white sail, surrounded by blue water and a white house in the background. | A picture of a man in a white shirt sits on the ground, his bare hand resting on his head. | A picture of a cozy living room with a floral chair and a gray chair placed on a hardwood floor, surrounded by a white wall a |
| | BLIP-Large | People standing around a man with a camera in a field. | There is a man that is sailing a pink boat in the water. | There is a man sitting on the ground with his feet crossed. | There is a chair that is sitting on the floor in the room. |
| | GIT | A group of people standing around a camera. | A man standing on a pink boat in a body of water. | A man sitting on the ground in front of a cave. | A chair sitting on top of a wooden floor next to a couch. |
| | Ours | A group of people stands in a field, with one man wearing a blue shirt and jeans holding a black camera. | A man sits on a pink sailboat with a blue and white sail, surrounded by blue water, with a white house in the background. | A man in a white shirt sits on the ground, his hand resting on his head as he gazes into the distance. | A cozy living room with a floral chair and a gray chair placed on a hardwood floor, surrounded by a white wall and a wooden door. |

*Figure 1.* Comparison of image captioning results from different models in the image shows that our SMM Transformer provides more accurate descriptions. Additionally, it captures color features well.

patible with sparse spike communication. Experimental results demonstrate that SMM Transformer achieves accuracy comparable to traditional ANNs in both visual tasks and multimodal tasks. We also designed ablation experiments to validate the contribution of each module to the overall performance. Our main contributions are summarized as follows:

- **Trainable spiking unit.** We propose PLMP, a parallel multistage LIF unit with learnable membrane time constants and thresholds, and introduce P-STBP to stably train PLMP-based deep spiking stacks.

- **Spike-driven token mixing.** We design SMSA, an attention-inspired token-mixing module built on PLMP read-out spikes. SMSA replaces dense pairwise softmax attention with channel-wise spike co-activation, sparse masking, and self-compensation.

- **Modality-aware fusion.** We introduce SMoE, which routes tokens to vision, language, or vision–language

experts to reduce cross-modal interference while keeping a shared interaction backbone.

- **Evaluation and analysis.** We validate SMM on visual and vision–language benchmarks with ablations, and provide an operator-level energy analysis of attention under the standard MAC/AC model.

## 2. Related Works

### 2.1. SNN Training Methods

Training strategies for SNNs are broadly classified into unsupervised and supervised paradigms. Unsupervised learning emulates biological synaptic plasticity, exemplified by Hebbian learning (Hebb, 2005) and spike-timing-dependent plasticity (STDP) (Caporale & Dan, 2008). Supervised techniques comprise (i) indirect ANN-to-SNN conversion (Sengupta et al., 2019; Yu et al., 2021), which often sacrifices accuracy, and (ii) direct gradient-based algorithms that

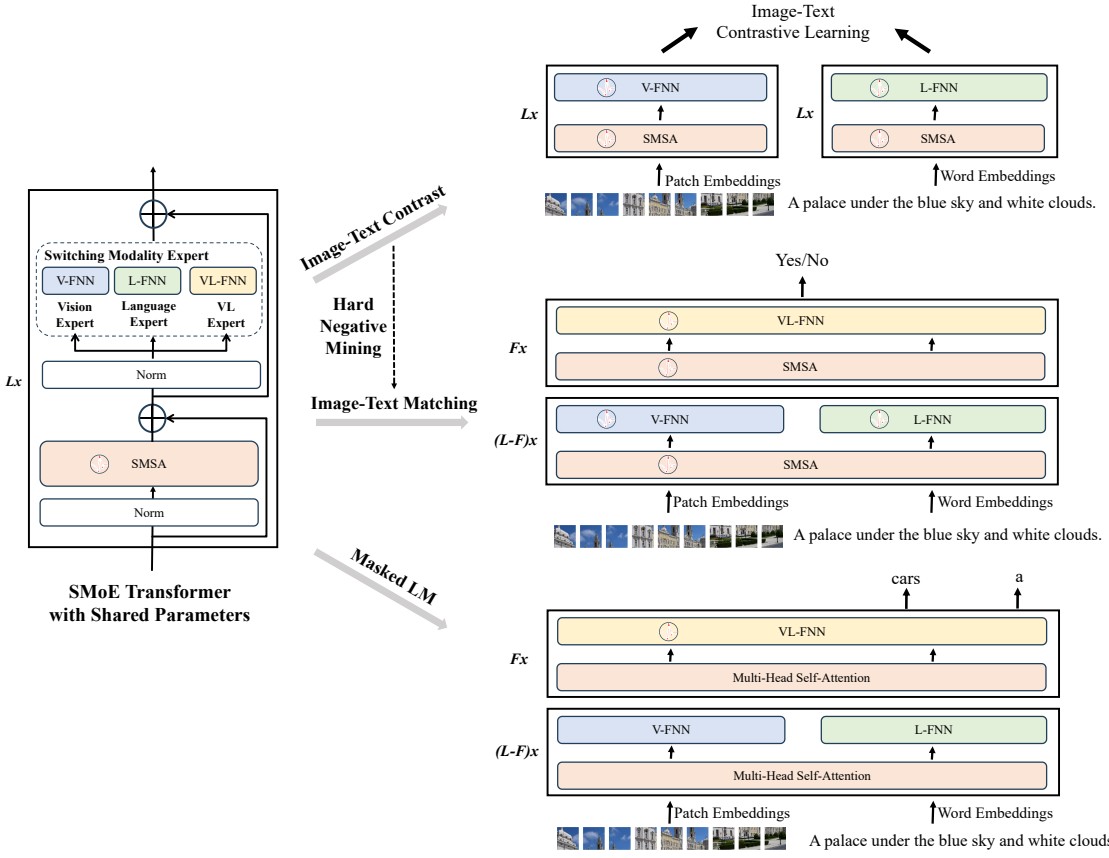

*Figure 2.* Overview of SMM Transformer. The model parameters are shared across image-text contrastive learning, masked language modeling, and image-text matching pre-training tasks.

circumvent spike non-differentiability, such as SpikeProp (Bohte et al., 2000) and STBP (Wu et al., 2018). Subsequent advances extend network depth (e.g., SEW-ResNet (Fang et al., 2021)) and integrate spiking blocks within Transformer frameworks for vision tasks (Zhou et al., 2022; Yao et al., 2024), yielding performance comparable to state-of-the-art CNNs (Jin et al., 2018; Zhang & Li, 2019; Kaiser et al., 2020; Rathi et al., 2020). Consequently, direct training has become the prevailing approach, promoting efficient end-to-end optimization.

## 2.2. Multimodal Tasks

### 2.2.1. IMAGE CAPTION

Image captioning has evolved from handcrafted descriptors—SIFT (Lowe, 1999) and HOG (Dalal & Triggs, 2005)—to CNN–RNN architectures endowed with spatial attention, exemplified by Show, Attend and Tell (Xu et al., 2015) and Bottom-Up and Top-Down Attention (Anderson et al., 2018). Transformer backbones (Vaswani et al., 2017) further strengthen relational reasoning—Object Re-

lation Transformer (Herdade et al., 2019) and X-Linear attention (Pan et al., 2020)—and enable fully detector-free, end-to-end pipelines (Wang et al., 2022). Complementary paradigms, including GAN-based training (Goodfellow et al., 2014), reinforcement learning (Chen et al., 2024), and dense captioning (Yang et al., 2017), alleviate exposure bias while enriching object semantics.

### 2.2.2. RETRIEVAL

Cross-modal retrieval seeks a shared embedding space for visual and textual modalities. In image-to-text retrieval, contrastive objectives align paired samples; attention or graph representations refine object-level correspondence (Xie et al., 2024), whereas attribute-centric descriptions bolster zero-shot generalisation (Zeng et al., 2024). Conversely, text-to-image retrieval employs dual-stream encoders (Suo et al., 2024) and diffusion priors (Koley et al., 2024) to capture nuanced semantics. Techniques such as knowledge injection (Suo et al., 2024), implicit concept mining (Wang et al., 2025), and multi-head hashing (Liu et al., 2024) further fortify robustness and scalability.

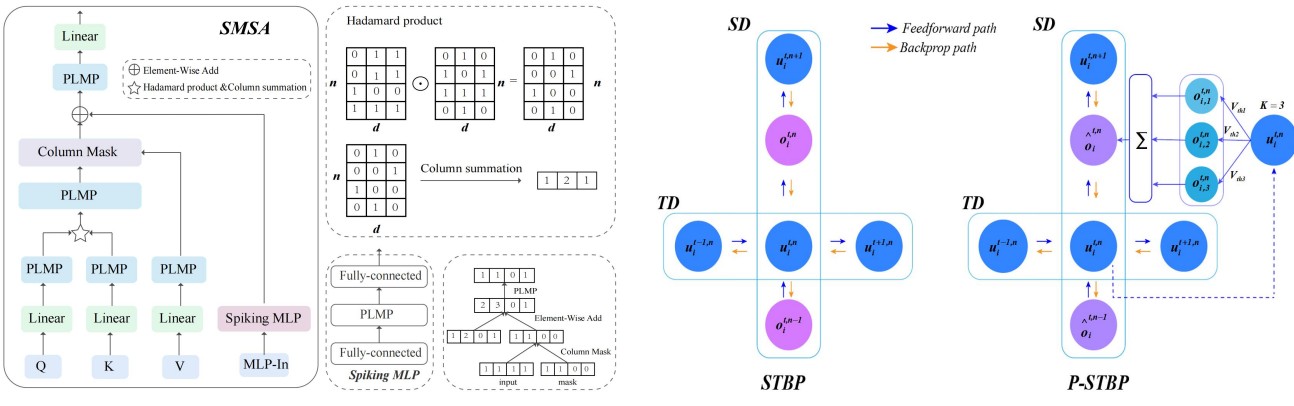

**(a). SMSA module**

**(b). P-STBP method**

*Figure 3.* Illustration of the proposed components. (a) SMSA first maps Query, Key, and Value into PLMP read-out spikes, then performs spike co-activation, column-wise aggregation, value masking, and self-compensation without constructing a dense $N \times N$ attention matrix. (b) P-STBP propagates gradients through both the temporal branch dynamics and the fixed-threshold PLMP read-out gate.

# 3. Method

As summarized in Fig. 2 and Fig. 3, SMM follows a Transformer-style residual stack, but replaces key building blocks to fit spiking computation. PLMP serves as the trainable spiking unit throughout the network, SMSA provides attention-inspired spike-driven token mixing without forming a dense $N \times N$ attention matrix, and SMoE injects modality-aware capacity in the feed-forward path while sharing a unified fusion backbone. We next describe each component and then detail how they are trained and evaluated in the multimodal pipeline.

## 3.1. PLMP

In conventional SNN training with LIF neurons, the membrane time constant ($\tau$) and voltage threshold ($V_{th}$) are treated as hyper-parameters that must be hand-tuned or optimised. Such manual specification is biologically implausible: $\tau$—the time for the membrane potential $u$ to equilibrate—should scale with membrane resistance ($R_m$) and capacitance ($C_m$). Additionally, the $V_{th}$, influenced by spiking input and discharge history, exhibits high variability both within and between cells. Computationally, forward propagation reduces to accumulating weighted presynaptic spikes and comparing the resulting membrane voltage with $V_{th}$. Consequently, empirical hyper-parameter selection inflates trial-and-error cost.

To overcome these limitations, we propose the PLMP neuron, whose temporal leakage and firing thresholds are learnable. Each PLMP unit contains $K$ parallel LIF branches. The $k$-th branch has its own membrane state $u_{i,k}^{t,n}$, learnable leakage factor $\alpha_k$, and learnable threshold $V_{th,k}$. Given the input current $r_i^{t,n}$, the branch dynamics are

$$u_{i,k}^{t,n} = \alpha_k u_{i,k}^{t-1,n}(1 - o_{i,k}^{t-1,n}) + r_i^{t,n}, \tag{1}$$

$$o_{i,k}^{t,n} = \mathcal{H}(u_{i,k}^{t,n} - V_{th,k}), \tag{2}$$

where $\mathcal{H}(\cdot)$ is the Heaviside spike function and $o_{i,k}^{t,n} \in \{0,1\}$ is the branch spike. The $K$ branch spikes are then aggregated into an internal spike count

$$c_i^{t,n} = \sum_{k=1}^{K} o_{i,k}^{t,n}. \tag{3}$$

PLMP propagates only a binary read-out spike to the next layer:

$$\hat{o}_i^{t,n} = \mathcal{H}(c_i^{t,n} - \theta_{\mathrm{ro}}), \quad \theta_{\mathrm{ro}} = K. \tag{4}$$

Thus, PLMP is not a stack of two unrelated LIF neurons. The parallel branches model heterogeneous temporal responses, while the fixed-threshold read-out gate performs coincidence-style binarization. Unless otherwise stated, the output of PLMP denotes the binary read-out spike $\hat{o}_i^{t,n}$ rather than the internal spike count $c_i^{t,n}$.

## 3.2. P-STBP

Given that PLMP units have different computational dynamics from standard LIF neurons, their forward propagation and back-propagation also differ. We therefore use P-STBP, a PLMP-specific extension of STBP tailored to multi-branch learnable membrane dynamics. Conceptually, P-STBP jointly propagates errors along spatial and temporal dimensions. As shown in Fig. 3(b), the spatial path transfers gradients across layers through PLMP read-out spikes, while the temporal path propagates gradients through the recurrent membrane states of parallel LIF branches.

For a connection from neuron $j$ in layer $n-1$ to neuron $i$ in layer $n$, the weight gradient accumulates the contributions

from all simulation steps and all PLMP branches:

$$\frac{\partial L}{\partial w_{ij}^n} = \sum_{t=1}^{T} \left( \sum_{k=1}^{K} \frac{\partial L}{\partial u_{i,k}^{t,n}} \frac{\partial u_{i,k}^{t,n}}{\partial r_i^{t,n}} \right) \frac{\partial r_i^{t,n}}{\partial w_{ij}^n}$$

$$= \sum_{t=1}^{T} \sum_{k=1}^{K} \frac{\partial L}{\partial u_{i,k}^{t,n}} \hat{o}_j^{t,n-1}, \qquad (5)$$

where $r_i^{t,n}$ is the input current of neuron $i$ at time step $t$, $w_{ij}^n$ is the synaptic weight, and $\hat{o}_j^{t,n-1}$ is the binary read-out spike from the previous layer. Similarly, the bias gradient is

$$\frac{\partial L}{\partial b_i^n} = \sum_{t=1}^{T} \sum_{k=1}^{K} \frac{\partial L}{\partial u_{i,k}^{t,n}}. \qquad (6)$$

The learnable leakage factor controls the contribution of the previous membrane state to the next membrane state. Its gradient is computed as

$$\frac{\partial L}{\partial \alpha_k} = \sum_{n,i,t} \frac{\partial L}{\partial u_{i,k}^{t+1,n}} u_{i,k}^{t,n} \left( 1 - o_{i,k}^{t,n} \right), \qquad (7)$$

where $\alpha_k$ is the learnable leakage factor of the $k$-th branch and $o_{i,k}^{t,n}$ is the corresponding branch spike.

The learnable threshold is updated through the surrogate derivative of the branch spike function:

$$\frac{\partial L}{\partial V_{th,k}} = -\sum_{n,i,t} \frac{\partial L}{\partial o_{i,k}^{t,n}} \phi_{br} \left( u_{i,k}^{t,n} - V_{th,k} \right), \qquad (8)$$

where $V_{th,k}$ is the threshold of the $k$-th branch and $\phi_{br}(\cdot)$ denotes the surrogate derivative used for the branch spike function.

These gradients show the core difference between P-STBP and standard STBP. Standard STBP propagates through a single membrane trajectory, whereas P-STBP aggregates gradients from $K$ parallel LIF branches and jointly optimizes synaptic weights, biases, leakage factors, and firing thresholds. Additional error-propagation details are provided in Appendix A.

### 3.3. SMSA

Building upon PLMP, we propose Spiking MLP Self-Attention (SMSA), an attention-inspired spike-driven token-mixing module that avoids constructing the dense $N \times N$ attention matrix. SMSA does not exactly reproduce softmax self-attention. Instead, it replaces pairwise token affinity with sequence-level channel-wise spike co-activation, which provides a practical accuracy-efficiency trade-off for SNN-based multimodal learning.

As illustrated in Fig. 3(a), we first project the input $X \in \mathbb{R}^{N \times D}$ and apply PLMP read-out neurons to obtain binary

spike tensors:

$$Q_s = \mathrm{PLMP}(XW^Q),$$
$$K_s = \mathrm{PLMP}(XW^K),$$
$$V_s = \mathrm{PLMP}(XW^V). \qquad (9)$$

where $Q_s, K_s, V_s \in \{0,1\}^{N \times D}$. Instead of computing $QK^\top$ followed by softmax, SMSA first estimates channel-wise spike co-activation:

$$G_c = SUM_c(Q_s \odot K_s), \qquad (10)$$

where $SUM_c(\cdot)$ aggregates over the token dimension and produces a $1 \times D$ channel vector. We then binarize this channel vector with PLMP to obtain a spike gate:

$$G_s = \mathrm{PLMP}(G_c). \qquad (11)$$

The gate is applied to the value spikes through column-wise masking:

$$M_s = G_s \otimes V_s. \qquad (12)$$

To compensate for information loss caused by spike binarization and the removal of dense token-pair interactions, we add a parallel self-compensation branch $SM(X)$ implemented as a spiking MLP:

$$Z_s = \mathrm{PLMP}(M_s \oplus SM(X)). \qquad (13)$$

Finally, the SMSA output is obtained by a linear output projection:

$$SMSA(X) = Z_s W^O. \qquad (14)$$

Here, $\odot$ denotes the Hadamard product, $\otimes$ denotes column-wise masking, and $\oplus$ denotes element-wise summation. The sparse mixing path operates on binary PLMP read-out spikes, while the final projection consumes the binary tensor $Z_s$.

Compared with vanilla self-attention, SMSA removes the $O(N^2)$ token-pair attention matrix and replaces it with channel-wise spike co-activation and masking. This design reduces arithmetic cost and enables sparse accumulate-based computation when spike activations are sparse. The trade-off is that SMSA no longer preserves the exact token-to-token inductive bias of dense attention. The self-compensation branch is therefore used to recover part of the lost expressiveness, which is empirically validated in Sec. 4.2.

$$L_R(\theta) = -E_{y_{1:t} \sim p_\theta}[r(y_{1:t})], \qquad (15)$$

$r(y_{1:t})$ is usually non-differentiable, the gradient of the $L_R(\theta)$ can be described by Eq.16.

$$\nabla L_R(\theta) \approx (r(y_s^{1:T}) - r(\hat{y}_{1:T})) \nabla \log p_\theta(y_s^{1:T}), \qquad (16)$$

where $y_s^{1:T}$ is a descriptive sample. $r(y_s^{1:T})$ is the greedy decoding score obtained from the current model.

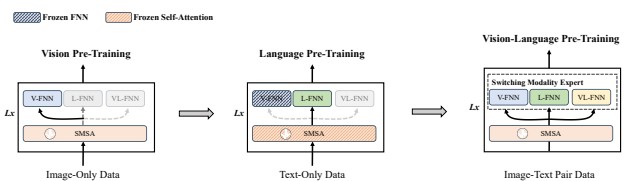

*Figure 4.* Illustration of stagewise pre-training strategy.

## 3.4. SMoE

In this section, we introduce SMoE, a spike-driven mixture-of-experts framework that dynamically selects modality-specific experts. SMoE is designed to preserve the representation benefit of expert routing while keeping the fusion pathway compatible with sparse spiking computation. Its structure is

$$y = \sum_{k=1}^{K} R_k(X_A) \cdot E_k(X_A), \quad (17)$$

where $X_A$ is the input sequence, $R_k(X_A)$ is the routing weight for the $k$-th expert, and $E_k(X_A)$ is the output of the corresponding spiking expert.

The routing mechanism selects the top-$K$ experts as

$$R(X_A) = \text{TopK}(\text{softmax}(X_A W_R)). \quad (18)$$

Because the routing is driven by spike-based representations, expert utilization is sparse and modality-aware. As shown in Fig. 2 and Fig. 4, we use a stagewise pre-training strategy. We first pretrain the vision expert and the shared SMSA module on image-only data. We then freeze these components and train the language expert with masked language modeling on text-only data. Finally, we perform vision-language pre-training to jointly optimize the full model on image-text pairs.

## 3.5. Image-Text Representations

### 3.5.1. IMAGE REPRESENTATIONS

Following vision Transformers (Dosovitskiy, 2020), the 2D image $\mathbf{v} \in \mathbb{R}^{H \times W \times C}$ is split and reshaped into $N = \frac{HW}{P^2}$ patches, where $(H, W)$ is the resolution of the input image, and $P$ is the patch resolution. The image patches are then flattened into vectors and linearly projected to obtain patch embeddings. We prepend a learnable special token $[T_{\text{CLS}}]$ to the sequence. Finally, the image input representations are obtained by summing the patch embeddings, learnable 1D position embeddings $V_{\text{pos}} \in \mathbb{R}^{(N+1) \times D}$, and image type embeddings $V_{\text{type}} \in \mathbb{R}^D$:

$$H_0^{\mathbf{v}} = [v_{[T_{\text{CLS}}]}, v_1^p, \dots, v_N^p] + V_{\text{pos}} + V_{\text{type}}, \quad (19)$$

where $H_0^{\mathbf{v}} \in \mathbb{R}^{(N+1) \times D}$ and the linear projection $V \in \mathbb{R}^{(P^2 C) \times D}$.

### 3.5.2. TEXT REPRESENTATIONS

Following BERT (Devlin et al., 2019), we tokenize the text to subword units using WordPiece. A start-of-sequence token $[T_{\text{CLS}}]$ and a special boundary token $[T_{\text{SEP}}]$ are added to the text sequence. Text input representations $H_0^{\mathbf{w}} \in \mathbb{R}^{(M+2) \times D}$ are computed by summing the corresponding word embeddings, text position embeddings, and text type embeddings:

$$H_0^{\mathbf{w}} = [w_{[T_{\text{CLS}}]}, w_1, \dots, w_M, w_{[T_{\text{SEP}}]}] + T_{\text{pos}} + T_{\text{type}}, \quad (20)$$

where $M$ indicates the length of tokenized subword units.

### 3.5.3. IMAGE-TEXT REPRESENTATIONS

Image and text input vectors are concatenated to form the image-text representation:

$$H_0^{\mathbf{vl}} = [H_0^{\mathbf{w}}, H_0^{\mathbf{v}}], \quad (21)$$

where $H_0^{\mathbf{vl}}$ denotes the joint image-text representation.

## 4. Experiments

### 4.1. Comparative Quantitative Experiments

#### 4.1.1. RETRIEVAL

We evaluate cross-modal retrieval on the COCO and Flickr30K datasets using the standard Karpathy split. SMM uses a dual-encoder retrieval architecture, enabling efficient image-text retrieval through dot-product similarity rather than exhaustive fusion-encoder scoring. Tab. 1 shows that SMM-Transformer-Base remains competitive among base-size models, while the Large variant further improves retrieval performance. We note that ALBEF-Base is stronger on some metrics, such as COCO R@1, because it uses a dense fusion encoder to rerank top-$k$ candidates. In contrast, SMM focuses on a spike-compatible retrieval pathway with sparse token mixing and efficient dual-encoder inference.

#### 4.1.2. IMAGE CAPTION

As shown in Tab. 2, SMM Transformer achieves competitive captioning quality while reducing estimated compute energy. In particular, SMM-Transformer-Large attains BLEU-1/2/3/4 of 83.50/65.56/56.48/45.33, METEOR 34.25, ROUGE-L 61.18, CIDEr 128.72, and SPICE 21.87, with an estimated compute energy of 22.1 mJ. We note that CIDEr/SPICE remain below the best ANN baseline, suggesting that spike binarization and linearized token mixing may trade some semantic consensus for efficiency. Fig. 1 further provides qualitative evidence that SMM captures fine-grained visual cues and generates semantically faithful captions.

*Table 1.* Fine-tuning results of text-retrieval (TR) and image-retrieval (IR) on COCO and Flickr30K. ALBEF first encodes images and text separately to obtain the top-k candidates, and then feed these representations into a fusion encoder to rerank the candidates. The others require to encode all image-text combinations by a fusion encoder.

| Model | MSCOCO (5K test set) | | | | | | Flickr30K (1K test set) | | | | | |
|---|---|---|---|---|---|---|---|---|---|---|---|---|
| | R@1 | R@5 | R@10 | R@1 | R@5 | R@10 | R@1 | R@5 | R@10 | R@1 | R@5 | R@10 |
| **Base-Size Models** | | | | | | | | | | | | |
| UNITER-Base | 64.4 | 87.4 | 93.1 | 50.3 | 78.5 | 87.2 | 85.9 | 97.1 | 98.8 | 72.5 | 92.4 | 96.1 |
| VILLA-Base | - | - | - | - | - | - | 86.6 | 97.9 | 99.2 | 74.7 | 92.9 | 95.8 |
| ViLT-Base | 61.5 | 86.3 | 92.7 | 42.7 | 72.9 | 83.1 | 83.5 | 96.7 | 98.6 | 64.4 | 88.7 | 93.8 |
| ALBEF-Base | 73.1 | 91.4 | 96.0 | 56.8 | 81.5 | 89.2 | 94.3 | 99.4 | 99.8 | 82.8 | 96.7 | 98.4 |
| SMM-Transformer-Base | 72.8 | 91.4 | 95.5 | 56.4 | 81.3 | 88.8 | 91.5 | 99.1 | 99.9 | 78.4 | 94.5 | 96.1 |
| **Large-Size Models** | | | | | | | | | | | | |
| UNITER-Large | 65.7 | 88.6 | 93.8 | 52.9 | 79.9 | 88.0 | 87.3 | 98.0 | 99.2 | 75.6 | 94.1 | 96.8 |
| VILLA-Large | - | - | - | - | - | - | 87.9 | 97.5 | 98.8 | 76.3 | 94.2 | 96.8 |
| SMM-Transformer-Large | **75.3** | **93.1** | **96.1** | **59.3** | **82.1** | **90.2** | **93.9** | **99.9** | **100.0** | **83.7** | **95.8** | **97.2** |

*Table 2.* Image captioning results on MSCOCO (Karpathy split). We report standard captioning metrics and measured power.

| Model | Type | BLEU-1 | BLEU-2 | BLEU-3 | BLEU-4 | METEOR | ROUGE-L | CIDEr | SPICE | Energy (mJ) |
|---|---|---|---|---|---|---|---|---|---|---|
| SCST | ANN | 78.1 | 61.9 | 47.0 | 35.2 | 27.0 | 56.3 | 114.7 | - | 102.3 |
| RFNet | ANN | 79.1 | - | - | 36.5 | 27.7 | 57.3 | 121.9 | 21.2 | 198.4 |
| Up-Down | ANN | 80.2 | 64.1 | 49.1 | 36.9 | 27.6 | 57.1 | 117.9 | 21.4 | 243.1 |
| GCN-LSTM | ANN | 80.8 | 65.5 | 50.8 | 38.7 | 28.5 | 58.5 | 125.3 | 22.0 | 323.9 |
| AoANet | ANN | 81.0 | 65.8 | 51.4 | 39.4 | 29.1 | 58.9 | 126.9 | 22.4 | 286.5 |
| X-LAN | ANN | 80.8 | - | - | 39.5 | 29.5 | 59.2 | 132.0 | 23.4 | 210.7 |
| X-Transformer | ANN | 81.9 | 66.9 | 52.4 | 40.3 | 29.6 | 59.5 | 131.1 | 23.4 | 368.4 |
| $M^2$ Transformer | ANN | 81.6 | 66.4 | 51.8 | 39.7 | 29.4 | 59.2 | 129.3 | 22.6 | 280.2 |
| RSTNet | ANN | 82.1 | 67.0 | 52.2 | 40.0 | 29.6 | 59.5 | 131.9 | 23.3 | 176.8 |
| GET | ANN | 81.6 | 66.5 | 51.9 | 39.7 | 29.4 | 59.1 | 130.3 | 22.8 | 150.6 |
| DLCT | ANN | 82.4 | 67.4 | 52.8 | 40.6 | 29.8 | 59.8 | 133.3 | 23.0 | 134.5 |
| PureT | ANN | 82.8 | 68.1 | 53.6 | 41.4 | 30.1 | 60.4 | 136.0 | 24.2 | 387.9 |
| SMM-Transformer-Large | SNN | 83.50 | 65.56 | 56.48 | 45.33 | 34.25 | 61.18 | 128.72 | 21.87 | **22.1** |

*Table 3.* Results on image classification and semantic segmentation.

| Models | Type | ImageNet (acc@1) | ADE20K (mIoU) |
|---|---|---|---|
| ViT-Base | ANN | 83.6 | - |
| BEiT-Base | ANN | 85.2 | 52.8 |
| PSSD | SNN | 66.8 | 29.1 |
| SMM-Base | SNN | 83.2 | 49.8 |

### 4.1.3. VISION TASKS

We evaluate SMM, employed as an image-only encoder, on two standard benchmarks: image classification (ImageNet) and semantic segmentation (ADE20K). As detailed in Table 3, SMM exhibits competitive performance and notably surpasses the SNN-based PSSD-Transformer. The image resolution for ImageNet is 224×224, and for ADE20K, it is 512×512. We perform intermediate fine-tuning on ImageNet-21k for all four models.

### 4.2. Ablation Study

**Pre-training tasks.** We conduct ablation studies to evaluate the impact of different vision-language pre-training tasks, with the results summarized in Tab. 4. Compared with the model trained only with image-text contrastive learning, adding masked language modeling and image-text matching consistently improves both NLVR2 and Flickr30K performance. These results show that SMM Transformer benefits from the same complementary supervision signals as ANN-based multimodal Transformers: contrastive learning improves global image-text alignment, masked language modeling strengthens textual grounding, and image-text matching with hard negatives improves fine-grained cross-modal discrimination.

**Modality-aware fusion.** We evaluate the contribution of SMoE in Tab. 4. Removing SMoE from the full SMM Transformer leads to consistent performance drops on both NLVR2 and Flickr30K, indicating that modality-aware expert routing is beneficial for vision-language fusion. This result suggests that a single shared feed-forward transformation is insufficient to fully handle the heterogeneous representations produced by image tokens, text tokens, and joint image-text tokens.

SMoE addresses this issue by allocating separate feed-forward capacity to vision, language, and vision-language representations. The vision and language experts pre-

*Table 4.* Ablation studies of SMM Transformer and vision-language pre-training tasks. "ITC" is short for image-text contrastive loss, "ITM" is image-text matching, and "MLM" is masked language modeling. The average of R@1, R@5 and R@10 is reported for Flickr30k. Results of NLVR2 are averaged over three runs.

| | Pre-Training Tasks | | | Transformer | | | NLVR2 | | Flickr30k | |
|---|---|---|---|---|---|---|---|---|---|---|
| | ITC | ITM | MLM | PLMP | SMSA | SMoE | dev | test-P | TR | IR |
| **1** | ✓ | ✗ | ✗ | ✓ | ✓ | ✓ | 54.74 | 56.18 | 90.49 | 82.24 |
| **2** | ✓ | ✗ | ✓ | ✓ | ✓ | ✓ | 72.53 | 70.56 | 91.60 | 84.29 |
| **3** | ✓ | ✓ | ✗ | ✓ | ✓ | ✓ | 73.63 | 72.44 | 92.46 | 82.53 |
| **4** | ✓ | ✓ | ✓ | ✗ | ✓ | ✓ | 76.74 | 76.89 | 90.72 | 84.27 |
| **5** | ✓ | ✓ | ✓ | ✓ | ✗ | ✓ | 77.45 | 76.46 | 91.67 | 83.46 |
| **6** | ✓ | ✓ | ✓ | ✓ | ✓ | ✗ | 77.55 | 76.89 | 93.04 | 85.53 |
| **7** | ✓ | ✓ | ✓ | ✓ | ✓ | ✓ | **78.36** | **77.53** | **94.61** | **86.22** |

*Table 5.* PLMP performance under different neuron configurations and temporal parameter initializations. The results compare static LIF, single-branch learnable LIF, and multi-branch PLMP variants on image captioning.

| Neuron configuration | BLEU-1 | BLEU-4 | METEOR | ROUGE-L | CIDEr | SPICE |
|---|---|---|---|---|---|---|
| LIF, $\alpha = 0.5$, $V_{th} = 0.6$ | 78.46 | 34.32 | 27.15 | 54.94 | 125.26 | 19.87 |
| PLMP, $K = 1$, $\alpha$ init. $= 0.5$, $V_{th} = 0.6$ | 78.89 | 36.00 | 27.30 | 55.98 | 125.69 | 20.77 |
| PLMP, $K = 3$, shared $\alpha$ init., $V_{th} = 0.3/0.6/0.9$ | 79.64 | 36.60 | 27.92 | 57.15 | 127.72 | 21.09 |
| PLMP, $K = 3$, multi-stage $\alpha$ init., $V_{th} = 0.3/0.6/0.9$ | 79.77 | 36.53 | 28.19 | 57.80 | 126.81 | 21.57 |

serve modality-specific transformations, while the vision-language expert provides additional capacity for joint cross-modal representations after spike-driven token mixing. In this way, SMoE complements SMSA: SMSA provides a shared sparse interaction pathway, whereas SMoE performs modality-aware nonlinear transformation in the feed-forward path. We further isolate the role of the shared SMSA pathway in Appendix B.1. Compared with separate SMSA parameters for image and text tokens in the early layers, shared SMSA achieves better NLVR2 and Flickr30K retrieval performance.

**PLMP and SMSA.** As shown in Tab. 4, experiments 4 and 5 validate the effectiveness of PLMP and SMSA, respectively. In experiment 4, we replace PLMP with standard LIF neurons and train the network with STBP. In experiment 5, we replace SMSA with the SSA baseline. Both replacements degrade performance, indicating that learnable multi-branch spiking dynamics and compensated spike-driven token mixing are important for the final model.

To further isolate the source of the PLMP gain, we report controlled neuron-configuration ablations in Tab. 5. The single-branch PLMP variant improves over the static LIF baseline while using the same branch count, showing that learnable leakage and threshold parameters already reduce the dependence on manually fixed LIF hyperparameters. The $K = 3$ PLMP variants further improve most captioning metrics, indicating that parallel branches provide richer temporal responses than a single membrane trajectory. The comparison between shared and multi-stage leakage initializations shows that initialization affects the training trajec-

*Table 6.* Expressiveness analysis of SMSA. Cosine similarity and JS/KL shift are computed with respect to the dense softmax self-attention control.

| Method | NLVR2 | F30K IR | Cos. | JS/KL |
|---|---|---|---|---|
| Dense Softmax-SA | 77.80 | 86.40 | 1.000 | 0.000 |
| SSA baseline | 76.46 | 83.46 | 0.882 | 0.064 |
| SMSA w/o comp. | 76.97 | 85.11 | 0.921 | 0.041 |
| SMSA full | 77.53 | 86.22 | 0.947 | 0.026 |

tory and final metrics moderately, but both variants remain consistently stronger than static LIF because the leakage factors and thresholds are adapted during optimization.

We further compare SMSA with dense softmax self-attention and intermediate spike-driven variants in Tab. 6. Dense softmax self-attention provides the strongest token-pair interaction, while SSA introduces a larger distribution shift. SMSA without self-compensation reduces this gap, and the full SMSA further improves both task performance and representation similarity to the dense control.

### 4.3. Energy Consumption

We evaluate efficiency from both module-level and whole-model perspectives. At the module level, we estimate the arithmetic energy of the attention block using the MAC/AC accounting model with 45nm CMOS constants, where $E_{MAC} = 4.6\,\text{pJ}$ and $E_{AC} = 0.9\,\text{pJ}$. For spike-driven operations, binary presynaptic activations convert multiply-accumulate operations into conditional accumulations, and inactive spikes skip the corresponding synaptic updates. We

*Table 7.* Operator-level arithmetic energy of one vanilla self-attention block and one SMSA block under the MAC/AC model. $E_{\mathrm{MAC}}$ and $E_{\mathrm{AC}}$ denote the energy costs of multiply-accumulate and accumulate operations, respectively.

| Type | Vanilla Self-Attention | SMSA |
|---|---|---|
| $Q, K, V$ projection | $E_{\mathrm{MAC}} \cdot 3ND^2$ | $E_{\mathrm{AC}} \cdot T \cdot R_{\mathrm{SA1}} \cdot 3ND^2$ |
| Attention mixing | $E_{\mathrm{MAC}} \cdot 2N^2D$ | $E_{\mathrm{AC}} \cdot T \cdot R_{\mathrm{SA2}} \cdot ND$ |
| Scale | $E_{\mathrm{MAC}} \cdot N^2$ | – |
| Softmax | $E_{\mathrm{MAC}} \cdot N^2$ | – |
| Output linear | $E_{\mathrm{MAC}} \cdot F_L$ | $E_{\mathrm{AC}} \cdot T \cdot R_L \cdot F_L$ |
| MLP layer 1 | – | $E_{\mathrm{AC}} \cdot T \cdot R_{\mathrm{M1}} \cdot F_{\mathrm{LM1}}$ |
| MLP layer 2 | – | $E_{\mathrm{AC}} \cdot T \cdot R_{\mathrm{M2}} \cdot F_{\mathrm{LM2}}$ |

*Table 8.* Empirical non-zero spike ratios used in the MAC/AC accounting. Lower values indicate stronger event sparsity.

| Split | $R_{\mathrm{SA1}}$ | $R_{\mathrm{SA2}}$ | $R_L$ | $R_{\mathrm{M1}}$ | $R_{\mathrm{M2}}$ |
|---|---|---|---|---|---|
| Train | 0.132 | 0.081 | 0.157 | 0.172 | 0.138 |
| Eval | 0.107 | 0.061 | 0.132 | 0.149 | 0.117 |

*Table 9.* Whole-model efficiency profile. Latency is measured under the same implementation and batch setting. Energy per pair is estimated from arithmetic operation counts and excludes memory access and hardware scheduling overhead.

| Metric | Dense | SMM | Change |
|---|---|---|---|
| Vision FLOPs (G) | 58.9 | 46.9 | -20.4% |
| Language FLOPs (G) | 42.1 | 34.9 | -17.1% |
| Whole-model FLOPs (G) | 170.4 | 127.6 | -25.1% |
| Latency / batch (ms) | 92.4 | 79.8 | -13.6% |
| Estimated energy / pair (mJ) | 356 | 279 | -21.6% |

therefore scale the AC cost by the measured non-zero spike ratio of each operator path.

Tab. 7 summarizes the operator-level accounting. Here, $N$ is the sequence length, $D$ is the hidden dimension, and $T$ is the number of SNN simulation steps. $F_L$ denotes the arithmetic count of the output linear projection, and $F_{\mathrm{LM1}}$ and $F_{\mathrm{LM2}}$ denote those of the two self-compensation MLP layers. $R_{\mathrm{SA1}}$, $R_{\mathrm{SA2}}$, $R_L$, $R_{\mathrm{M1}}$, and $R_{\mathrm{M2}}$ are the measured non-zero spike ratios of the corresponding operator paths. For the scale and softmax terms, we use one scalar arithmetic operation per attention logit as a proxy, without modeling hardware-specific exponential or division units.

As shown in Tab. 8, SMSA maintains low activation rates across projection, mixing, output, and compensation paths. Using these measured ratios, SMSA reduces the estimated operator-level arithmetic energy of the attention block by up to 97% compared with vanilla self-attention under the same hidden size, sequence length, and simulation-step setting. This estimate reflects attention-block arithmetic cost only and does not include memory access, scheduling overhead, or hardware-specific kernel effects.

We further report whole-model FLOPs, latency, and arithmetic energy estimates in Tab. 9. The end-to-end gains are smaller than the attention-block gains because FFN/SMoE, embedding, projection, and task-head computations still occupy a substantial fraction of the total computation. Thus, the module-level estimate demonstrates the arithmetic advantage of SMSA, while the whole-model profile gives a more conservative view of SMM efficiency.

## 5. Conclusion

This work introduces an SNN-based framework for multimodal learning. The proposed PLMP neuron, together with P-STBP, stabilizes learnable multi-branch spiking dynamics. SMSA replaces dense pairwise softmax attention with attention-inspired channel-wise spike co-activation and self-compensation, providing a practical accuracy-efficiency trade-off. SMoE further introduces modality-aware expert routing for vision-language fusion. Experiments across visual and multimodal tasks show that SMM Transformer achieves competitive accuracy compared with ANN baselines. Our analysis shows large operator-level arithmetic savings for the attention module and more moderate but consistent whole-model efficiency gains. Future work will study memory-aware deployment and real-device neuromorphic measurements.

## Acknowledgments

This work was partly supported by Ningbo Key R&D Program (2025Z047) , Ningbo Major Application Demonstration Program (2025Z199) and Ningbo Youth Science and Technology Innovation Leading Talent Project (2024QL044).

## Impact Statement

This paper presents work whose goal is to advance the field of machine learning. There are many potential societal consequences of our work, none of which we feel must be specifically highlighted here.

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

## A. Derivations for PLMP and P-STBP

This section provides the error-propagation details omitted from the main text. The forward dynamics of PLMP, including branch membrane updates, branch spikes, internal spike-count aggregation, and binary read-out spikes, have already been defined in Sec. 2.

P-STBP uses surrogate derivatives for the non-differentiable branch spike and read-out spike functions:

$$\frac{\partial o_{i,k}^{t,n}}{\partial u_{i,k}^{t,n}} \approx \phi_{\mathrm{br}}\left(u_{i,k}^{t,n} - V_{th,k}\right), \quad \frac{\partial \hat{o}_i^{t,n}}{\partial c_i^{t,n}} \approx \phi_{\mathrm{ro}}\left(c_i^{t,n} - \theta_{\mathrm{ro}}\right). \tag{22}$$

Here, $\phi_{\mathrm{br}}(\cdot)$ is used for branch spikes, while $\phi_{\mathrm{ro}}(\cdot)$ is used for the fixed-threshold PLMP read-out gate. The task-specific loss provides the upstream gradient at the output layer. For hidden PLMP layers, the read-out spike receives gradients from the next layer:

$$\begin{aligned}
\frac{\partial L}{\partial \hat{o}_i^{t,n}} &= \sum_{j \in \mathcal{N}_{n+1}} \sum_{k=1}^{K} \frac{\partial L}{\partial u_{j,k}^{t,n+1}} w_{ji}^{n+1}, \\
\frac{\partial L}{\partial o_{i,k}^{t,n}} &= \frac{\partial L}{\partial \hat{o}_i^{t,n}} \phi_{\mathrm{ro}}\left(c_i^{t,n} - \theta_{\mathrm{ro}}\right) - \frac{\partial L}{\partial u_{i,k}^{t+1,n}} \alpha_k u_{i,k}^{t,n}, \\
\frac{\partial L}{\partial u_{i,k}^{t,n}} &= \frac{\partial L}{\partial o_{i,k}^{t,n}} \phi_{\mathrm{br}}\left(u_{i,k}^{t,n} - V_{th,k}\right) + \frac{\partial L}{\partial u_{i,k}^{t+1,n}} \alpha_k \left(1 - o_{i,k}^{t,n}\right).
\end{aligned} \tag{23}$$

The first line corresponds to spatial error propagation across layers, where $\mathcal{N}_{n+1}$ denotes the set of postsynaptic neurons in the next layer. The second line contains two paths for the branch spike: the read-out path at the current time step and the reset path that affects the next membrane state. The third line combines the current branch-spike surrogate gradient and the temporal recurrent gradient from the next simulation step.

Since the read-out spike $\hat{o}_i^{t,n}$ is generated from the internal spike count $c_i^{t,n}$, gradients must first pass through the read-out gate and then be distributed to the $K$ branch spikes. Meanwhile, each branch spike also affects the next membrane state through the reset term in the PLMP membrane update. Therefore, P-STBP propagates errors not only across layers and time steps, but also across the internal branch aggregation structure of PLMP.

If the learnable leakage factor is represented by an unconstrained parameter $m_k$, we use $\alpha_k = \mathrm{sigmoid}(m_k)$ and apply the chain rule:

$$\frac{\partial L}{\partial m_k} = \frac{\partial L}{\partial \alpha_k} \alpha_k \left(1 - \alpha_k\right). \tag{24}$$

This parameterization keeps the leakage factor in a stable range during training. These equations clarify why P-STBP differs from standard STBP. Standard STBP propagates through a single LIF membrane trajectory, whereas P-STBP must propagate through the temporal membrane recursion, the branch spike surrogate, the internal spike-count aggregation, and the binary read-out gate. This allows PLMP to learn branch-specific leakage factors and thresholds while propagating only binary read-out spikes to subsequent layers.

## B. Additional Experiments and Statistics

This section provides additional sensitivity results on the number of PLMP branches and further analyzes the shared SMSA design.

*Table 10.* Sensitivity of PLMP to the number of parallel LIF branches. Cost is normalized by the static LIF baseline.

| Neuron configuration | BLEU-4 | CIDEr | Cost |
|---|---|---|---|
| LIF | 34.32 | 125.26 | 1.00× |
| Learnable-LIF | 35.84 | 126.18 | 1.06× |
| PLMP, $K = 2$ | 36.34 | 126.94 | 1.12× |
| PLMP, $K = 3$ | 36.60 | 127.72 | 1.17× |
| PLMP, $K = 4$ | 36.56 | 127.63 | 1.24× |

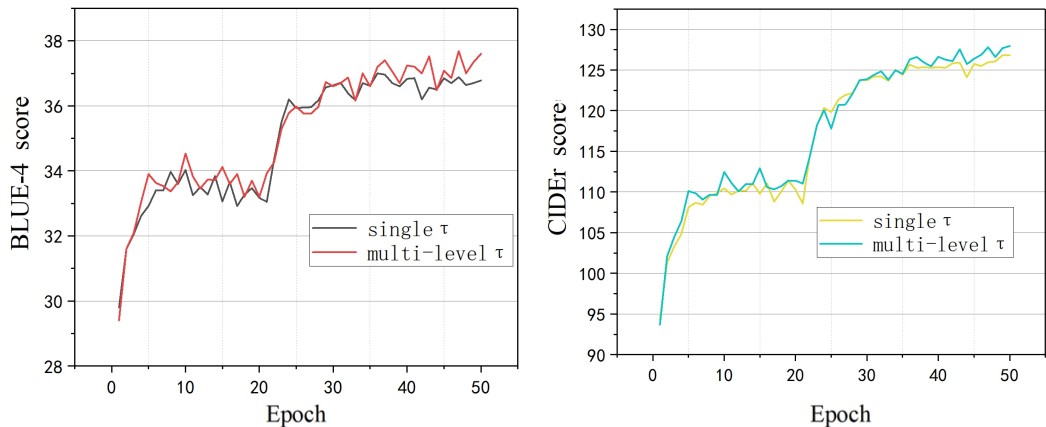

*Figure 5.* Training curves for PLMP variants with different temporal parameter initializations.

*Table 11.* Controlled ablation of shared SMSA in the early multimodal layers. Separate SMSA uses different spike-driven token-mixing parameters for image patches and text tokens in the first $L - F$ layers, while shared SMSA uses a common SMSA pathway before modality-aware SMoE transformations.

| Setting | NLVR2 dev | NLVR2 test-P | Flickr30K TR | Flickr30K IR |
|---|---|---|---|---|
| Separate SMSA | 77.53 | 76.58 | 93.21 | 85.98 |
| Shared SMSA + SMoE | 79.17 | 79.42 | 94.57 | 86.33 |

Tab. 10 studies the sensitivity of PLMP to the number of parallel branches. Increasing $K$ from 1 to 3 consistently improves BLEU-4 and CIDEr, indicating that multiple temporal branches are useful for representing diverse firing dynamics. However, increasing $K$ from 3 to 4 brings negligible performance improvement while increasing computational cost. Therefore, we use $K = 3$ by default in the main experiments, as it provides the best accuracy-cost trade-off.

Fig. 5 compares PLMP variants with different temporal parameter initializations. The curves show that learnable temporal parameters reduce sensitivity to the initial leakage configuration. Although different initializations lead to slightly different early training trajectories, the final performance remains close after optimization. This supports the motivation of PLMP: branch-specific learnable temporal parameters allow the neuron to adapt its firing dynamics during training instead of relying on a manually fixed membrane constant.

### B.1. Shared SMSA versus Separate SMSA

Tab. 11 studies whether image and text tokens should use separate SMSA parameters or a shared SMSA pathway in the early layers. The separate variant encodes image patches and text tokens with different SMSA parameters before fusion, whereas the shared variant exposes both modalities to a common spike-driven token-mixing process. The shared design improves both NLVR2 and Flickr30K retrieval, suggesting that early common SMSA helps visual and textual tokens form a more aligned representation space before the modality-aware SMoE feed-forward transformations. This ablation complements Tab. 6. The expressiveness analysis studies how self-compensation reduces the representation gap between SMSA and dense softmax self-attention, while the shared-SMSA ablation studies whether a common early token-mixing pathway is beneficial for cross-modal alignment.

