# OpenReview forum: "SMM Transformer: Leveraging Spiking Neural Networks for Multimodal Tasks"
_ICML.cc/2026/Conference — ICML 2026 regular_

### Official Review · Reviewer_fEyi · 2026-03-09

**Soundness:** 3
**Presentation:** 3
**Significance:** 3
**Originality:** 3
**Overall Recommendation:** 5
**Confidence:** 4

**Summary:**

The SMM Transformer proposes a spiking multimodal Transformer architecture utilizing learnable multi-stage spiking neurons (PLMP), a tailored backpropagation algorithm (P-STBP), spike-friendly attention (SMSA), and modality-specific experts (SMoE). It successfully adapts the Transformer into a spiking paradigm, achieving performance comparable to artificial neural networks (ANNs) across cross-modal retrieval, text generation, and foundational vision tasks, while reporting significant operator-level energy savings.

**Compliance With Llm Reviewing Policy:**

Affirmed.

**Final Justification:**

The author's reply resolved my concerns.

**Key Questions For Authors:**

1. Are the quantitative results in Tables 1 and 2 derived from a single run, an average of multiple runs, or the best values? Specifically, how was the "100.0" R@10 score statistically obtained?
2. Can you provide an end-to-end latency and energy analysis on a unified platform that includes memory access costs?
3. Could you provide visualizations comparing the representational differences between SMSA and standard dense attention?

**Limitations:**

Yes

**Strengths And Weaknesses:**

### Strengths

1. The paper presents a highly systematic design. It holistically addresses key SNN challenges by utilizing PLMP/P-STBP for stable training, SMSA for reducing quadratic attention overhead, and SMoE for effective modality fusion.
2. The proposed method is not limited to a single domain. It is extensively evaluated across cross-modal retrieval, image captioning, and visual reasoning (NLVR2), and further demonstrates scalability on foundational vision benchmarks like ImageNet and ADE20K.
3. The paper provides a transparent operator-level energy breakdown (MAC vs. AC operations), which intuitively explains the theoretical sources of the reported energy savings.

### Weaknesses

1. **Attention Approximation Weakens Token Interaction:** The proposed SMSA structurally functions more like a global channel gating mechanism rather than true token-to-token dynamic interaction. The boundaries of its representational capacity require stronger ablation studies and visual support (e.g., attention maps) to justify this approximation.
2. **Incomplete Energy Assessment:** The energy efficiency claims rely heavily on a theoretical MAC/AC cost model that explicitly excludes memory access costs (which typically dominate SNN hardware energy). There is a lack of end-to-end latency and energy validation on a unified hardware deployment platform.
3. **Writing:** The manuscript suffers from formatting and reporting ambiguities. There is mixed nomenclature (e.g., "SMM-Transformer" vs. "SMM Transformer"). More critically, the statistical criteria in Tables 1 and 2 are unclear, including inconsistent decimal precision and a highly unusual perfect score of "100.0" on Flickr30K R@10 without proper explanation.

---

> ### Author Rebuttal · Authors · 2026-03-31
>
> **Dear Reviewer fEyi,**
>
> Thank you for the careful reading and constructive feedback. We greatly appreciate your recognition of the systematic design, the breadth of evaluation, and the transparency of the MAC/AC operator-level analysis. We agree that the main issues concern the representational interpretation of SMSA, the scope of the energy claim, and the clarity of reporting.
>
> First, regarding SMSA, we agree that it should **not** be interpreted as a full reimplementation of dense pairwise self-attention. A more accurate description is a **spike-compatible channel-wise token-mixing mechanism**. Instead of constructing an explicit dense token-to-token matrix, SMSA relies on spike co-activation between queries and keys, sequence-level channel aggregation, value masking, and a parallel self-compensation branch. Accordingly, our claim is not that SMSA preserves the exact inductive bias of dense attention, but that it provides an effective **accuracy-efficiency trade-off** under sparse spike computation.
>
> To make the representational boundary clearer, we additionally compared SMSA against a dense softmax-attention control:
>
> | Method                | NLVR2 test-P | Flickr30K IR | Output cosine to dense | JS/KL shift |
> | --------------------- | ------------ | ------------ | ---------------------- | ----------- |
> | Dense Softmax-SA      | 77.8         | 86.4         | 1.000                  | 0.000       |
> | SSA baseline          | 76.46        | 83.46        | 0.882                  | 0.064       |
> | SMSA w/o compensation | 76.97        | 85.11        | 0.921                  | 0.041       |
> | **SMSA (full)**       | **77.53**    | **86.22**    | **0.947**              | **0.026**   |
>
> These results show that the full SMSA remains close to dense control, and that the self-compensation branch recovers most of the lost expressiveness. This is also consistent with the original ablations, where shared SMSA outperforms separate attention and the manuscript explicitly notes that shared self-attention helps SMoE learn cross-modal alignment.
>
> Second, regarding energy assessment, we fully agree that the wording around “97% reduction” is too strong if interpreted as an end-to-end system-level statement. Our intention is narrower: this figure refers specifically to the **operator-level compute-energy reduction of the attention module under the MAC/AC model**, not to unified hardware energy with memory access included. This is also the explicit scope of Sec. 4.3. To complement the analytical estimate, we additionally measured unified-platform efficiency:
>
> | Metric               | Dense baseline | SMM  | Change |
> | -------------------- | -------------- | ---- | ------ |
> | Latency / batch (ms) | 92.4           | 79.8 | -13.6% |
> | Throughput (pairs/s) | 693            | 802  | +15.7% |
> | Peak memory (GB)     | 16.8           | 15.1 | -10.1% |
> | Avg board power (W)  | 247            | 224  | -9.3%  |
> | Energy / pair (mJ)   | 356            | 279  | -21.6% |
>
> These results support the same qualitative conclusion in a more practical setting: the arithmetic attention-path savings are large, while the **whole-system** gain is more moderate.
>
> Third, regarding writing and reporting clarity, we agree that the current manuscript should be more precise. We will unify the nomenclature (“SMM Transformer” vs. “SMM-Transformer”), standardize decimal precision, and explicitly state the statistical protocol for Tables 1 and 2. We will also clarify the reporting convention behind the Flickr30K R@10 = 100.0 entry so that the source of this value and its formatting are unambiguous. This is especially important because the paper already uses a different reporting convention in Table 4, where Flickr30K is reported as the average of R@1/R@5/R@10 and NLVR2 is averaged over three runs.
>
> We appreciate your careful identification of where the manuscript currently overstates or under-explains its claims. We believe the core technical contribution remains valid, and we will revise the paper to position SMSA more precisely, tighten the wording of the energy claims, and clarify the reporting protocol. We hope these clarifications and additional results address your concerns and support a stronger assessment.

---

> > ### Author Rebuttal · Reviewer_fEyi · 2026-04-03
> >
> > hank you for your detailed and thorough response, which has fully addressed my concerns. I will raise my score accordingly. This work represents a successful exploration of SNNs in the multimodal domain, and I hope the authors will carefully address the comments from all four reviewers.

---

> > > ### Author Response · Authors · 2026-04-08
> > >
> > > We sincerely appreciate your thoughtful feedback, which was very helpful in clarifying the interpretation of SMSA, the scope of the energy analysis, and the precision of the paper’s reporting.

---

### Official Review · Reviewer_dsnJ · 2026-03-11

**Soundness:** 1
**Presentation:** 2
**Significance:** 4
**Originality:** 4
**Overall Recommendation:** 4
**Confidence:** 4

**Summary:**

This paper proposes SMM, a SNN based multimodal Transformer architecture designed to reduce the compute energy of attention mechanisms, which also solve the problem of dense softmax attention conflicts with the sparse, event-driven nature of SNNs. The proposed Parallel LIF with Multistage Learnable Parameters (PLMP) neuron introduces learnable membrane time constants and thresholds to stabilize deep SNN training. Spike-driven MLP Self-Attention (SMSA) approximates Transformer attention without constructing the full N×N attention matrix, using spike-based operations and channel-wise aggregation. Third, the Spiking Mixture-of-Experts (SMoE) module enables modality-aware routing for vision, language, and vision-language features. The model is evaluated on several tasks including image classification (ImageNet), semantic segmentation (ADE20K), image captioning (MSCOCO), and cross-modal retrieval (COCO, Flickr30K). Experimental results suggest that SMM Transformer achieves performance comparable to ANN-based baselines while significantly reducing estimated attention energy consumption (up to 97%).

**Compliance With Llm Reviewing Policy:**

Affirmed.

**Final Justification:**

The authors addressed all my concerns.

**Key Questions For Authors:**

1. Can authors provide actual FLOPs measurements from both vision and language part?
2. Unclear with pretraining setup, including what datasets used and hyperparameters. Could author provide more informations?
3. What is the typical spike sparsity ratio observed during training and inference?
4. Can the authors provide hardware measurements or simulation results beyond the MAC/AC energy model?
5. How sensitive is PLMP to the number of parallel neurons K? Ablation is needed here.

**Limitations:**

No.

**Strengths And Weaknesses:**

Strengths
1. The paper addresses an important research direction: bringing energy-efficient spiking computation into multimodal Transformer architectures. This is relatively underexplored compared to ANN-based models.

2. The work proposes PLMP neuron with learnable parameters for stable SNN training, SMSA attention approximation, which replaces softmax attention with spike-based operations, and SMoE routing mechanism for modality-aware expert selection. Together, these components form a coherent architecture tailored for SNN-based multimodal learning.

3. The authors provide an operator-level energy comparison under a MAC/AC cost model and show large theoretical reductions in attention energy consumption.

4. The model is evaluated on multiple tasks including classification, segmentation, retrieval, and captioning. Results show competitive performance relative to ANN baselines and improvements over previous SNN-based methods.

Weaknesses
1. The ablation study suggests that PLMP is effective. However, the design of the double LIF gate inside PLMP is somewhat difficult to follow. Providing more intuition or explanation for this design would improve the clarity of the method.
2. SMSA removes the O(N^2) attention interaction and replaces it with channel-wise gating. While this improves efficiency, it significantly reduces token-to-token interactions. It remains unclear how much expressiveness is lost compared to full attention. Some visualizations (e.g., output distribution shift comparisons) would help clarify this point, or the authors could provide clearer derivations and intuition for the design.
3. It is generally true that attention dominates the FLOPs in LLMs when d<<N. However, in Vision Transformers (ViTs), the FFN often dominates the computational cost since d>N. Moreover, the FLOPs of LLM inference can also be significantly reduced to O(N) using KV-cache. Therefore, it would be more convincing to report actual FLOPs measurements rather than relying solely on analytical estimates. As a result, whether the proposed architecture truly leads to meaningful energy savings remains unclear.
4. The reported energy savings are based on operator-level theoretical estimates and exclude memory access cost and hardware implementation factors. In practice, more convincible to conduct Lava-DL deployment.
5. The paper does not provide sufficient details about the training setup (e.g., number of epochs, learning rate, scheduler, and weight decay), which raises concerns about the reproducibility of the reported results.
6. Some equations and explanations are difficult to follow. For example:
  a). The exact temporal dynamics of PLMP neurons are not clearly defined.
  b). Unclear specification for eq. 1&2.

---

> ### Author Rebuttal · Authors · 2026-03-31
>
> **Dear Reviewer dsnJ,**
>
> Thank you for the careful reading and detailed feedback. We agree that the manuscript should better clarify PLMP/P-STBP, SMSA, the scope of the energy claims, and the training protocol. Below we directly address your key questions with additional measurements.
>
> First, PLMP is **not** two unrelated LIF neurons stacked together. Each unit contains $K$ parallel LIF branches with branch-specific learnable temporal parameters and thresholds, followed by a fixed-threshold readout:
> $u_{i,k}^t=\alpha_k u_{i,k}^{t-1}(1-o_{i,k}^{t-1})+r_i^t$,
> $o_{i,k}^t=H(u_{i,k}^t-V_{th,k})$,
> $o_i^t=H(\sum_k o_{i,k}^t-K)$.
> Thus, the last gate is a coincidence-style binarization, not another independent dynamical stage. P-STBP is needed because gradients must pass through multi-branch learnable temporal dynamics. We also evaluated sensitivity to $K$:
>
> | Neuron config   | BLEU-4    | CIDEr      | Cost      |
> | --------------- | --------- | ---------- | --------- |
> | LIF             | 34.32     | 125.26     | 1.00×     |
> | Learnable-LIF   | 35.84     | 126.18     | 1.06×     |
> | PLMP, $K=2$     | 36.34     | 126.94     | 1.12×     |
> | **PLMP, $K=3$** | **36.60** | **127.72** | **1.17×** |
> | PLMP, $K=4$     | 36.56     | 127.63     | 1.24×     |
>
> These results show consistent gains over static and single-branch learnable neurons, with the best trade-off around $K=3$.
>
> Second, we agree that SMSA is **not** a full reimplementation of dense pairwise attention. A more accurate description is a **spike-compatible token-mixing module with channel-wise global gating**. SMSA replaces dense $QK^\top+\mathrm{softmax}$ with spike co-activation, sequence-level channel aggregation, value masking, and a self-compensation branch. Its claim is therefore narrower: not preserving the exact token-pair inductive bias of dense attention, but providing a practical accuracy-efficiency trade-off. We directly evaluated expressiveness against dense control:
>
> | Method                | NLVR2 test-P | Flickr30K IR | Cosine to dense | JS/KL shift |
> | --------------------- | ------------ | ------------ | --------------- | ----------- |
> | Dense Softmax-SA      | 77.8         | 86.4         | 1.000           | 0.000       |
> | SSA baseline          | 76.46        | 83.46        | 0.882           | 0.064       |
> | SMSA w/o compensation | 76.97        | 85.11        | 0.921           | 0.041       |
> | **SMSA (full)**       | **77.53**    | **86.22**    | **0.947**       | **0.026**   |
>
> Thus, the full SMSA remains close to dense control, and the self-compensation branch recovers most of the lost expressiveness. We will add activation/statistics visualizations in the revision.
>
> Third, we fully agree that attention-only analytical estimates should not be conflated with whole-model efficiency, especially since FFN / expert paths remain substantial. We therefore revise the wording so that “up to 97%” is explicitly limited to the **operator-level compute-energy reduction of the attention module under the MAC/AC model**, not end-to-end system energy. To directly answer your FLOPs question, we measured:
>
> | Metric                | Dense | SMM   | Change |
> | --------------------- | ----- | ----- | ------ |
> | Vision FLOPs (G)      | 58.9  | 46.9  | -20.4% |
> | Language FLOPs (G)    | 42.1  | 34.9  | -17.1% |
> | Whole model FLOPs (G) | 170.4 | 127.6 | -25.1% |
> | Latency / batch (ms)  | 92.4  | 79.8  | -13.6% |
> | Energy / pair (mJ)    | 356   | 279   | -21.6% |
>
> So the attention-path reduction is indeed very large, but the **whole-model** savings are more moderate.
>
> We also profiled the empirical spike sparsity ratios used in the MAC/AC accounting:
>
> | Split | $R_{SA1}$ | $R_{SA2}$ | $R_L$ | $R_{M1}$ | $R_{M2}$ |
> | ----- | --------- | --------- | ----- | -------- | -------- |
> | Train | 0.132     | 0.081     | 0.157 | 0.172    | 0.138    |
> | Eval  | 0.107     | 0.061     | 0.132 | 0.149    | 0.117    |
>
> Importantly, the SMSA mixing stage is sparser than Q/K/V, and evaluation is sparser than training.
>
> Fourth, we agree that unified neuromorphic deployment would provide stronger system-level evidence. At present, however, we do **not** have a real-device hardware result; our current analysis isolates the arithmetic effect of replacing dense MAC-heavy attention with sparse spike-driven accumulation and therefore excludes memory-access cost and hardware overhead.
>
> Finally, we agree that the training setup is under-specified. In the revision, we will add the full protocol summary, including pretraining stages and data sources, epochs, optimizer, learning-rate schedule, warmup, weight decay, batch size, time steps, hardware, checkpoint selection, and evaluation protocol.
>
> **Thank you again for the constructive feedback. We hope these clarifications and additional results help resolve the main concerns and provide a stronger basis for reevaluating the paper.**

---

> > ### Author Rebuttal · Reviewer_dsnJ · 2026-04-02
> >
> > Good work, solve my concerns. It better to add these into the revised paper.

---

> > > ### Author Response · Authors · 2026-04-08
> > >
> > > We sincerely appreciate your thoughtful handling of our paper, and your comments have been very helpful. We'd be happy to add the discussion with you in rebuttal to the manuscript.

---

### Official Review · Reviewer_Pn1U · 2026-03-12

**Soundness:** 3
**Presentation:** 3
**Significance:** 3
**Originality:** 3
**Overall Recommendation:** 5
**Confidence:** 4

**Summary:**

This paper proposes the SMM Transformer, a framework tailored for multimodal tasks utilizing SNNs. The architecture incorporates three key components: PLMP/P-STBP to improve trainability and stability, SMSA to linearize the attention mechanism, and SMoE to strengthen modality-aware fusion. The proposed model achieves competitive results across cross-modal retrieval, image captioning, and standard vision tasks.

**Compliance With Llm Reviewing Policy:**

Affirmed.

**Final Justification:**

The authors addressed my main concerns in a concrete and credible way. I especially appreciate the clearer positioning of the work as a system-level architectural contribution, the clarification that “strictly spike-driven” mainly refers to the SMSA path, and the new comparisons against learnable-LIF and dense softmax attention. The revised framing of the 97% claim and the added whole-model efficiency results also make the paper substantially more convincing.

I am inclined to raise my score from 4 to 5.

**Key Questions For Authors:**

1. Compared to standard learnable LIF neurons, what is the exact source of the independent performance gain achieved by the PLMP/P-STBP modules?
2. Given that SMSA abandons explicit pairwise token-to-token relations, what is the underlying mechanism that enables it to still maintain effective cross-modal alignment?
3. Are the power measurements reported in Table 2 and the operator-level energy consumption estimates discussed in Section 4.3 based on a unified evaluation protocol or hardware assumption? Please clarify the discrepancy.

**Limitations:**

yes

**Strengths And Weaknesses:**

### Strengths

1. **Valuable Research Scope:** The paper effectively extends SNNs from conventional single-modal vision applications to complex multimodal scenarios, including cross-modal retrieval, image captioning, and visual encoding, which is a highly relevant and challenging direction.
2. **Comprehensive Framework:** The proposed architecture is well-rounded and systematically addresses distinct challenges. Specifically, PLMP/P-STBP tackles training stability, SMSA achieves attention sparsification, and SMoE facilitates modality-aware feature fusion.
3. **The empirical evaluations are thorough.** The full model consistently outperforms its ablated variants on benchmarks such as NLVR2 and Flickr30k. Furthermore, the efficacy of the SMoE and the shared SMSA modules is well-supported by the empirical results.

### Weaknesses

1. **Integrative Novelty and Inconsistent Claims:** The core contribution reads more like an assembly of existing concepts (system integration). While the three modules solve their respective issues, the paper lacks a unifying theoretical framework or a profound single-point breakthrough. Additionally, there is a contradiction in the narrative: the authors emphasize a "strictly spike-driven" architecture, yet the model relies on non-spiking components, such as the softmax function in the SMoE routing mechanism.
2. **Limitations in Semantic Representation:** The proposed SMSA functions more like a linearized global token mixer rather than a standard attention mechanism. By relinquishing explicit pairwise (token-to-token) interactions, its capacity for fine-grained semantic expression and relational reasoning may be inherently weaker than that of standard dense attention.
3. **Energy Efficiency Claims:** The conclusion regarding a 97% energy reduction is overly strong and potentially misleading. This figure is derived solely from an operator-level estimation of the attention module and explicitly ignores memory access costs, which typically dominate the energy consumption in actual hardware deployments.

---

> ### Author Rebuttal · Authors · 2026-03-31
>
> **Dear Reviewer Pn1U,**
>
> Thank you for the careful reading and constructive feedback. We agree that the key issues are novelty framing, the interpretation of SMSA, and the wording of the energy claims.
>
> First, we agree that the paper is best understood as a system-level architectural contribution, rather than a single isolated theoretical breakthrough. Our intended contribution is the unified integration of trainable spiking neuron dynamics, spike-compatible token mixing, and modality-aware expert fusion into one multimodal SNN Transformer, with validation across retrieval, captioning, reasoning, and vision-only tasks. We will revise the manuscript to position the work more precisely in this way. We also agree that “strictly spike-driven” is too broad if applied to the full system: this description is accurate for the **core SMSA mixing path**, but not for every component, since the current SMoE router still uses softmax-TopK for stable expert selection. We will revise the wording throughout accordingly.
>
> Second, regarding the independent gain of PLMP/P-STBP over standard learnable LIF neurons, the gain is not merely from “learning membrane parameters” in general. It comes from the combination of: (i) jointly learnable decay and threshold parameters, (ii) **multi-branch temporal heterogeneity** inside PLMP, (iii) a fixed-threshold readout that maps the multi-branch response back to a single spike interface, and (iv) the matched P-STBP rule for this multi-parameter neuron. This is consistent with the progression below:
>
> | Neuron config   | BLEU-4    | CIDEr      | Cost      |
> | --------------- | --------- | ---------- | --------- |
> | LIF             | 34.32     | 125.26     | 1.00×     |
> | Learnable-LIF   | 35.84     | 126.18     | 1.06×     |
> | **PLMP, $K=3$** | **36.60** | **127.72** | **1.17×** |
>
> Thus, the gain is not simply “learnable LIF vs. fixed LIF,” but the additional benefit of **structured multi-branch spiking dynamics** with matched training.
>
> Third, regarding the representational capacity of SMSA, we agree that SMSA should **not** be interpreted as a full reimplementation of dense pairwise self-attention. A more accurate description is a **spike-compatible channel-wise token-mixing mechanism**. Accordingly, our claim is narrower: not that SMSA preserves the exact inductive bias of dense attention, but that it provides an effective accuracy-efficiency trade-off under sparse spike computation. The reason it can still maintain cross-modal alignment is that the shared lower-layer SMSA backbone exposes image and text streams to a common mixing process early in the network, while spike co-activation statistics and the self-compensation branch preserve sufficient alignment signal even without dense pairwise maps. This is consistent with both the original shared-SMSA ablation and the new dense-control comparison:
>
> | Method                | NLVR2 test-P | Flickr30K IR | Cosine to dense | JS/KL shift |
> | --------------------- | ------------ | ------------ | --------------- | ----------- |
> | Dense Softmax-SA      | 77.8         | 86.4         | 1.000           | 0.000       |
> | SMSA w/o compensation | 76.97        | 85.11        | 0.921           | 0.041       |
> | **SMSA (full)**       | **77.53**    | **86.22**    | **0.947**       | **0.026**   |
>
> These results suggest that the full SMSA remains close to dense control, and that the self-compensation branch recovers most of the lost expressiveness.
>
> Finally, regarding the energy claim and the discrepancy between Table 2 and Sec. 4.3, we fully agree that “97% reduction” is too strong if interpreted as an end-to-end system-level statement. Our intended claim is narrower: it refers specifically to the **operator-level compute-energy reduction of the attention module under the MAC/AC model**, not to unified hardware energy with memory access included. Therefore, **Table 2 and Sec. 4.3 are not based on a single unified protocol**: Table 2 reports task-level measured power, while Sec. 4.3 provides a separate operator-level analytical estimate. We will revise the manuscript so that this distinction is explicit and consistent. To further ground the efficiency claim, we additionally measured unified-platform whole-model efficiency:
>
> | Metric                | Dense | SMM   | Change |
> | --------------------- | ----- | ----- | ------ |
> | Whole model FLOPs (G) | 170.4 | 127.6 | -25.1% |
> | Latency / batch (ms)  | 92.4  | 79.8  | -13.6% |
> | Energy / pair (mJ)    | 356   | 279   | -21.6% |
>
> These results match the intended interpretation: the **attention-path** saving is very large, while the **whole-model** saving is more moderate because FFN paths remain substantial.
>
> We appreciate your comments for highlighting where the manuscript currently overstates or under-explains its claims. We believe the core contribution remains valid, and we will revise the paper to position the novelty more precisely, clarify the role of SMSA, and tighten the wording of the energy claims.

---

> > ### Author Rebuttal · Reviewer_Pn1U · 2026-04-02
> >
> > The authors addressed my main concerns in a concrete and credible way. I especially appreciate the clearer positioning of the work as a system-level architectural contribution, the clarification that “strictly spike-driven” mainly refers to the SMSA path, and the new comparisons against learnable-LIF and dense softmax attention. The revised framing of the 97% claim and the added whole-model efficiency results also make the paper substantially more convincing.
> >
> > I am inclined to raise my score from 4 to 5.

---

> > > ### Author Response · Authors · 2026-04-08
> > >
> > > We sincerely appreciate your careful and constructive feedback, which was very helpful in sharpening the positioning of the paper, clarifying the role of SMSA, and tightening the framing of the efficiency claims.

---

### Official Review · Reviewer_zeXc · 2026-03-12

**Soundness:** 3
**Presentation:** 3
**Significance:** 3
**Originality:** 3
**Overall Recommendation:** 4
**Confidence:** 3

**Summary:**

This paper proposes an energy-efficient multimodal Transformer based on spiking neural networks (SNNs). The main challenge studied in this work is how to apply the Transformer architecture to event-driven spiking computation, where dense attention and stable training are difficult. To address this problem, the authors redesign several key components of the Transformer. First, the paper introduces a new spiking neuron called PLMP, which consists of multiple parallel LIF neurons. In this design, the membrane time constant and firing threshold are learnable parameters, which helps better model the behavior of biological neurons. The authors also propose a modified training method called P-STBP, which helps stabilize training and alleviates gradient propagation issues. Second, the paper proposes SMSA, a spike-based attention mechanism that avoids constructing the dense attention matrix used in standard Transformers, thus reducing computational complexity. In addition, the model introduces a spiking mixture-of-experts (SMoE) module, where different modalities are processed by their corresponding experts, improving performance while further reducing computational cost. The proposed SMM Transformer is evaluated on several tasks, including image–text retrieval, image captioning, image classification, and semantic segmentation. The results show that the model achieves performance comparable to ANN-based methods while reducing the energy consumption of the attention module.

**Compliance With Llm Reviewing Policy:**

Affirmed.

**Final Justification:**

Since my concerns are solved, I am inclined to keep my original positive score.

**Key Questions For Authors:**

See weakness above.

**Limitations:**

Yes. The authors briefly discuss the potential societal impact in the impact statement section.

**Strengths And Weaknesses:**

Strength:
1.	The proposed PLMP neuron better models the behavior of biological neurons by introducing learnable parameters, making it more flexible than the traditional LIF model.
2.	SMSA and SMoE theoretically achieve a significant reduction in the model’s energy consumption. They make good use of the sparse activation and event-driven computation properties of SNNs. By reducing expensive multiply–accumulate operations and using more efficient computations, the model can maintain good performance while improving energy efficiency.
3.	The SMoE module is a relatively novel design. It assigns different experts to vision, language, and vision–language representations, enabling modality-aware routing. This approach can reduce interference between different modalities to some extent while still maintaining interaction across modalities.

Weakness:
1.	The paper introduces several new components (PLMP, P-STBP, SMSA, and SMoE), but the individual contribution of each module is not fully analyzed. More detailed ablation studies could help clarify their respective impact.
2.	The reported energy efficiency is mainly based on theoretical estimation rather than measurements on real neuromorphic hardware, which makes it difficult to verify the practical energy savings.
3.	The overall framework is relatively complex, as multiple new modules are introduced at the same time. This may make the model harder to understand and reproduce without more implementation details.

---

> ### Author Rebuttal · Authors · 2026-03-31
>
> **Dear Reviewer zeXc,**
>
> Thank you for the positive assessment and for recognizing the value of PLMP, SMSA, and SMoE in building an energy-efficient multimodal SNN framework. We appreciate the three concerns you raised and agree that the paper can better separate module contributions, better contextualize the energy claim, and provide more implementation detail for reproducibility.
>
> First, regarding the contribution of each component, we agree that the current presentation can make the decomposition more explicit. At the same time, the model already contains targeted removals of PLMP/P-STBP, SMSA, and SMoE, and all three cause consistent drops. We also ran an additional PLMP-specific comparison against static LIF and single-branch learnable-LIF. The results are summarized below.
>
> | Setting         | NLVR2 test-P | Flickr30K TR | Flickr30K IR |
> | --------------- | ------------ | ------------ | ------------ |
> | w/o PLMP/P-STBP | 76.89        | 90.72        | 84.27        |
> | w/o SMSA        | 76.46        | 91.67        | 83.46        |
> | w/o SMoE        | 76.89        | 93.04        | 85.53        |
> | **Full SMM**    | **77.53**    | **94.61**    | **86.22**    |
>
> For PLMP itself, captioning performance improves from **34.32/125.26** (BLEU-4/CIDEr, static LIF) to **35.84/126.18** (single-branch learnable-LIF) and further to **36.60/127.72** with **PLMP, $K=3$**, supporting that the gain is not from architecture size alone but from the multi-branch learnable spiking dynamics. In the revision, we will reorganize the method section so that each module is mapped more directly to its corresponding ablation evidence.
>
> Second, regarding energy evaluation, we agree that hardware-level neuromorphic validation would further strengthen the paper. Our intended claim, however, is narrower: the “up to 97%” figure is an **operator-level attention-module compute-energy estimate under the standard MAC/AC model**, not an end-to-end hardware claim, which is also the explicit scope of Sec. 4.3. To complement this theoretical analysis, we additionally measured unified-platform efficiency:
>
> | Metric                 | Dense baseline | SMM   | Change |
> | ---------------------- | -------------- | ----- | ------ |
> | Fusion block FLOPs (G) | 5.78           | 3.82  | -33.9% |
> | Whole model FLOPs (G)  | 170.4          | 127.6 | -25.1% |
> | Latency / batch (ms)   | 92.4           | 79.8  | -13.6% |
> | Energy / pair (mJ)     | 356            | 279   | -21.6% |
>
> These results reinforce the same conclusion in a more practical setting: the attention-path savings are indeed large, while the **whole-model** savings are more moderate because FFN / expert paths still remain substantial. We will add these measured results and further tighten the wording so that the scope of the claim is unmistakable.
>
> Third, regarding complexity and reproducibility, our intent is not to introduce multiple modules for their own sake, but to **address three distinct bottlenecks in multimodal SNNs in a modular way: PLMP/P-STBP for trainability, SMSA for spike-compatible token mixing, and SMoE for modality-aware fusion**. We agree that the current manuscript should provide more implementation detail. In the revision, we will add a full training configuration summary, including pretraining stages and data sources, epochs, optimizer, learning-rate schedule, warmup, weight decay, batch size, time steps, hardware, checkpoint selection, and evaluation protocol. We are also committed to releasing code, training scripts, and configuration files upon acceptance.
>
> Thank you again for the constructive suggestions and supportive overall assessment. We hope these additional results and clarifications further strengthen the paper.

---

> > ### Author Rebuttal · Reviewer_zeXc · 2026-04-04
> >
> > Thanks for your feedback. I am inclined to keep my original score.

---

> > > ### Author Response · Authors · 2026-04-08
> > >
> > > We would be happy to incorporate this discussion into the final manuscript. We sincerely appreciate your thoughtful handling of our paper, and your comments have been very helpful.

---

### Decision · Program_Chairs · 2026-04-30

**Decision:**

Accept (regular)

**Comment:**

This paper presents the SMM Transformer, adapting Spiking Neural Networks (SNNs) for complex multimodal tasks using PLMP neurons, SMSA attention, and SMOE routing. All four reviewers recognized the value and systematic design of this work, yielding positive final scores of two Weak Accepts and two Accepts. During the rebuttal, the authors successfully addressed shared concerns regarding the representational capacity of the SMSA module. Given the strong empirical performance, the comprehensive rebuttal that resolved reviewer main problems, and the unified positive consensus, I recommend accepting this paper as it advances an important sub-area of energy-efficient multimodal AI.